# Obsessive–Compulsive Symptoms in Children Are Related to Sensory Sensitivity and to Seeking Proxies for Internal States

**DOI:** 10.3390/brainsci13101463

**Published:** 2023-10-16

**Authors:** Ilil Tal, Matti Cervin, Nira Liberman, Reuven Dar

**Affiliations:** 1School of Psychological Sciences, Tel Aviv University, Tel Aviv 69978, Israel; ilil2798@gmail.com (I.T.); niralib@tauex.tau.ac.il (N.L.); 2Department of Clinical Sciences, Lund University, 221 00 Lund, Sweden; matti.cervin@med.lu.se

**Keywords:** obsessive–compulsive disorder (OCD), compulsive behavior, development, sensory sensitivity, proxies for internal states, SPIS model, children

## Abstract

Symptoms of obsessive–compulsive disorder are related to atypical sensory processing, particularly sensory over-responsivity, in both children and adults. In adults, obsessive–compulsive symptoms are also associated with the attenuation of access to the internal state and compensatory reliance on proxies for these states, including fixed rules and rituals. We aimed to examine the associations between sensory over-responsivity, the tendency to seek proxies for internal states, and obsessive–compulsive symptoms in children. Parents of 404 children between 5 and 10 years of age completed online measures of obsessive–compulsive symptoms, seeking proxies for internal states, sensory over-responsivity, and anxiety. Linear regression, dominance analysis, and network analysis were used to explore the unique associations between these variables. The tendency to seek proxies for internal states was more strongly associated with obsessive–compulsive symptoms than with anxiety symptoms and uniquely associated with all major obsessive–compulsive symptom dimensions except obsessing. Both the tendency to seek proxies for internal states and sensory over-responsivity were significantly associated with obsessive–compulsive symptoms, but the association was significantly stronger for the tendency to seek proxies for internal states. While limited by the sole reliance on the parent-report, the present study shows that the tendency to seek proxies for internal states could help clarify the developmental processes involved in the onset of obsessive–compulsive symptoms during childhood and that sensory sensitivity may be important to consider in this process.

## 1. Introduction

Obsessive–compulsive disorder (OCD) is defined via the presence of obsessions (recurrent and persistent thoughts, urges, or images that the individual attempts to ignore, suppress, or neutralize) and/or compulsions (repetitive behaviors or mental acts that the individual feels driven to perform [1]). Most cases of OCD have their onset prior to adulthood and the prevalence of pediatric OCD is estimated to range between 1–3% [2]. Variability in symptoms, as well as in their severity, course, and treatment responsiveness, suggests heterogeneity in pediatric OCD [3,4]. For example, whereas some children and adolescents with OCD present with symptoms triggered by intrusive thoughts and anxiety (e.g., repetitive lock checking triggered by intrusive thoughts of burglary), others present with symptoms that are primarily triggered by sensorimotor stimuli [5], where perceptions, sensations, or urges precede repetitive behaviors [6,7]. Generally, the symptoms of OCD were found to cluster around four major symptom dimensions: (1) disturbing thoughts and checking, (2) obsessions of a sexual or religious nature (often referred to as taboo thoughts), (3) symmetry and ordering, and (4) contamination and cleaning [8], and these dimensions are present in both youth and adults [3]. Treatments for pediatric OCD include serotonin reuptake inhibitors and cognitive behavioral therapy with exposure and response prevention [9]. While these treatments are clearly more efficacious than the control conditions, around 30–50% of those treated do not benefit sufficiently [9], and it is increasingly recognized that a better understanding of the mechanisms involved in the onset and maintenance of pediatric OCD is needed to improve and tailor treatment.

The present study extends a recent model of OCD, titled the Seeking Proxies for Internal States (SPIS) model [10]. According to the SPIS model, OCD is characterized by impaired access to the internal states, which drives people with OCD to seek and rely on “proxies” for these states (see Figure 1 for the process hypothesized via the SPIS model). Internal states in this model are defined broadly, encompassing emotions and preferences as well as bodily states and sensations, whereas proxies comprise relatively verifiable indices of internal states. For example, an individual with OCD may infer how much they love their partner (an internal state) based on how much money they have spent on buying them presents (the proxy), or decide that they are hungry if their shirt is hanging loose on their body. While the SPIS model has been substantiated by considerable empirical evidence (for recent reviews see [10,11]), there has not been an attempt to apply the model to children. The present study constitutes the first step in this direction. We reasoned that examining the SPIS model in children will add to our understanding of the relevance of the model across different age groups, as well as suggest avenues for understanding early difficulties in accessing the internal states.

The present study also aims to explore whether atypical sensory processing is involved in the relation between the access to the internal states and obsessive–compulsive symptoms. Preliminary findings from research with both children and adults with OCD suggest multiple connections between obsessive–compulsive symptoms and atypical sensory processing, sensory over-responsivity, and sensory hypersensitivity [5,12,13,14,15,16,17,18,19,20,21]. Sensory processing refers to the way in which the nervous system receives, organizes, and deciphers sensory stimuli from internal and external sources [22]. Sensory over-responsivity (SOR), also referred to as sensory sensitivity, specifically entails difficulty ignoring or over-reacting to sensory stimuli [23]. In other terms, SOR is considered to reflect an imbalance between the sensitivity to stimuli and the habituation to them [24,25]. A recent web-based study concluded that SOR is a transdiagnostic phenomenon linked to all the major symptom dimensions of OCD as well as to OCD-related difficulties such as skin picking, hair pulling, hoarding, and body dysmorphic symptoms [22]. In a large twin study of children and adolescents, Van Hulle and colleagues [19] also found a relationship between SOR, as reported by parents, and OCD symptoms. Finally, SOR was shown to be a very common feature in a sample of 86 youth with OCD and correlated significantly with all the major symptom dimensions of OCD in this sample [14].

Research of the clinical manifestations of atypical sensory processing has often relied on Dunn’s theoretical model [26,27], which classifies sensory processing and responsiveness along two axes—the response threshold of the nervous system (high/low) and the strategy of response (accordance/counteract). This classification results in four groups: “low registration” (high/accordance); “sensory seeking” (high/counteract); “sensory sensitivity” (low/accordance); and “sensory avoiding” (low/counteract). A previous study utilizing this model found that adults with OCD reported higher levels of sensory sensitivity, sensation avoidance, low registration, and lower levels of sensory seeking compared to a non-clinical control group [28]. Regarding pediatric OCD specifically, Hazen and colleagues [29] described six cases in which the intolerance of ordinary sensory experiences (e.g., the sensation created by socks, common household smells, and everyday sounds made by family members) were reported as primary obsessive–compulsive symptoms. In the cases described, children performed compulsions to relieve sensations or to relieve the stress resulting from the aversive sensations; importantly, these compulsions occurred without the precursory presence of intrusive thoughts or obsessions.

Common manifestations of sensory intolerance in children include the aversion to tactile stimuli (e.g., specific fabrics or textures), intense reactions when touched by others (e.g., hugs from parents, haircuts), extreme sensitivity to visual or auditory stimuli (e.g., bright lights, sirens), or the refusal to eat certain foods (based on texture, odor, or temperature). Dar and colleagues (Ref. [16], Study 1) found that such strong reactions to everyday sensory events were related to childhood ritualism among pre-school age children, as reported by their parents, even after controlling for the children’s levels of anxiety. In a follow-up study (Ref. [16], Study 2), scores on a scale measuring oral and tactile hypersensitivity (OTHS), which was based on the first study and which we adopt in the current study as well, were related to obsessive–compulsive symptoms in adult participants.

In the present study, we explore the possibility that the tendency to seek proxies for internal states may be involved in linking sensory sensitivity and obsessive–compulsive symptoms, particularly compulsive behaviors. Specifically, social developmental theories suggest that learning to identify and label one’s own internal states requires empathic sharing with and reflection of one’s experiences by other people [30,31,32]. One implication of these theories is that if a child’s subjective experiences are often not shared by others, then this process might be disrupted. Children with sensory over-responsivity, who experience a wide range of stimuli as aversive, live in a social situation in which their inner experiences may not be readily shared by others. As a result, learning to correctly access and label one’s internal states is likely to be impaired, which, according to the SPIS model, would lead to seeking and relying on proxies for these states, including fixed rules and compulsive rituals. 

As sensory sensitivity may be related not only to OCD in children but also to anxiety [14], we included a measure of anxiety symptoms in the study. The main objectives of the study were to examine (1) whether the tendency to seek proxies for internal states is linked to obsessive–compulsive symptoms also in children and (2) what role sensory sensitivity may play in the link between the tendency to seek proxies for internal states and obsessive–compulsive symptoms. We had three hypotheses. First, in line with the SPIS model, we expected that the tendency to seek proxies for internal states would be moderately to strongly associated with obsessive–compulsive symptoms and more strongly so than with anxiety symptoms. Second, given their potential roles in the development of obsessive–compulsive symptoms, we expected that both the tendency to seek proxies for internal states and sensory sensitivity would be uniquely associated with obsessive–compulsive symptoms. Third, because we hypothesize that the tendency to seek proxies for internal states may act as a link between sensory sensitivity and obsessive–compulsive symptoms, we expected that the association between seeking proxies for internal states and obsessive–compulsive symptoms would be stronger than the association between sensory sensitivity and obsessive–compulsive symptoms. For the latter, because we only had access to cross-sectional data, we did not conduct a mediation analysis [33]. Instead, we considered the study to be a first exploration of how the tendency to seek proxies for internal states, sensory sensitivity, and obsessive–compulsive symptoms in young children are interconnected.

## 2. Method

### 2.1. Participants and Procedure

Four hundred and four (*N* = 404) Jewish Hebrew-speaking parents (175 fathers and 229 mothers) of children (189 boys and 215 girls) between 5 and 10 years of age (*M* = 7.60, *SD* = 1.33) participated in an online study. The participants were registered users of an Israeli internet database and received a small monetary reward (4.5 NIS; approximately $1.5) for their participation. 

Participants could participate in the survey if they had at least one child between 5 and 10 years of age; if they had more than one child in this age range, they were instructed to choose one child and respond to the entire survey with regard to that child. Parents were first asked to provide general information about their child’s age, gender, medical and developmental history, and then proceeded to respond to questionnaires assessing their child’s obsessive–compulsive symptoms, proxy-seeking behavior, sensory sensitivity, and anxiety levels. The study protocol was approved by the Research Ethics Council of Tel-Aviv University.

### 2.2. Measures

Obsessive–compulsive symptoms were measured with the Obsessive–Compulsive Inventory—Child Version (OCI-CV; [34]). The OCI-CV comprises 21 items scored on a 3-point Likert-type scale (0 = never, 1 = sometimes, and 2 = always), which are divided into six subscales that represent different dimensions of OCD: doubting/checking (five items), obsessing (four items), and washing, hoarding, ordering, and neutralizing (three items each). In previous studies, the OCI-CV was found to have high internal consistency (Cronbach’s α ≥ 0.81) for the total score and subscales [34], and good validity, test-retest reliability, and internal consistency in both clinical and non-clinical samples [35]. In the current study, we used a Hebrew version of the OCI-CV, which was approved by the Israeli Ministry of Health. We adapted the scale to a 5-point Likert-type scale to maximize the consistency with the response scales of other questionnaires used in this study. We also examined the three subscales of the OCI-CV that best correspond to the major symptom dimensions of pediatric OCD, namely the obsessing, ordering, and washing subscales [36], as well as the doubting/checking subscale because this may be of particular relevance to the tendency to seek proxies for internal states. The internal consistency of the full OCI-CV scale in the current sample was excellent (Cronbach’s α = 0.92) and the internal consistency of the subscales was adequate: obsessing (α = 0.80), ordering (α = 0.87), washing (α = 0.77), and doubting/checking (α = 0.78).

Seeking proxies for internal states was measured with the Seeking Proxies for Internal States—Child Version (SPISI-CV), a scale developed for this study based on the Seeking Proxies for Internal States Inventory for adults (SPISI; [37]). The scale comprises 21 questions regarding different proxy-seeking behaviors that children might use, such as “chooses what to eat based on fixed pre-determined criteria” and “seeking confirmation from others for their attitudes and opinions”. Parents were asked to rate the extent to which these behaviors characterized their children on a 5-point scale, ranging from 1 (to a small extent) to 5 (very much so). The internal consistency of the SPISI-CV in the current sample was excellent (Cronbach’s α = 0.92). Notably, while relying on parents’ reports is clearly not identical to obtaining direct reports from the children, this approach was successfully applied in previous research [16,19]. To minimize bias, we chose the items so that they tap observable behaviors and do not require inferences about the children’s internal states. 

Oral and tactile sensitivity was measured with the Oral and Tactile Hypersensitivity Scale (OTHS; [16]). The OTHS combines unique items from the Sensory Profile assessment [38] to form an internally consistent scale (Cronbach’s α = 0.89) that focuses on oral and tactile hypersensitivities. Due to the length of the entire survey, we created a shortened version of the OTHS scale comprising 12 items scored on a 5-point scale ranging from 1 (to a small extent) to 5 (very much so). The internal consistency of the shortened version of the questionnaire in the current sample was high (Cronbach’s α = 0.86).

Trait anxiety was measured with the short version of the Screen for Child Anxiety Related Emotional Disorders (SCARED; [39]). The original scale consists of 38 items that assess five factors: panic/somatic anxiety, generalized anxiety, separation anxiety, social phobia, and school phobia. Responders are asked to assess the intensity and frequency of these phenomena in their children on a 3-point scale (0 = not true/seldom true, 1 = sometimes true, 2 = true/often true). The short 5-item version that we used in the present study has similar psychometric properties as the full SCARED [40]. The internal consistency of the short version in the current sample was good (Cronbach’s α = 0.67).

### 2.3. Statistical Analyses

Zero-order associations between the variables of the study were examined using Pearson correlation coefficients. We also examined associations between child gender and child age and each of the major variables of the study using Pearson correlations (age) and independent sample *t*-tests (gender). To examine the unique contributions of seeking proxies for internal states and sensory sensitivity to obsessive–compulsive symptoms, two statistical models were used. First, we conducted linear regression followed by dominance analysis. In the linear regression, OCD symptom scores on the OCI-CV were entered as the dependent variable, and age, gender, the tendency to seek proxies for internal states, sensory sensitivity, and anxiety symptoms were entered as predictors. We followed up the linear regression with dominance analysis, in which all possible subset models of independent variables were tested to examine the degree of unique variation in the dependent variable accounted for by each independent variable [41]. The linear regressions were also conducted separately for boys and girls and for mother- and father-reports.

Next, we estimated a network model that included obsessive–compulsive symptoms, the tendency to seek proxies for internal states, sensory sensitivity, and anxiety symptoms. In a network model, the unique association between each variable pair is estimated by accounting for all linear associations among the full set of variables. To facilitate interpretation, we illustrated the network using a network graph where each variable is represented as a node (a circle) and each unique association by an edge (a line). Only associations that were statistically significant were plotted and the placement of each node was determined according to the Fruchterman–Reingold layout which places nodes with many and strong connections centrally, while node pairs with a strong connection are placed closely [42]. Network edges were estimated using the R library *BGGM*. Unique associations were examined with partial correlations. All variables were treated as continuous and 95% credible intervals (CI) were used to control the familywise Type I error. 

The network model was used to test our three hypotheses. We carried this out by using 5000 posterior estimates for each edge and then examining whether one edge was significantly larger than another. We used posterior probabilities to test the hypotheses and posterior probabilities above 95% were considered to indicate a statistically significant difference in line with the hypothesis. For exploratory purposes, we also used a network model that included four subscales of the OCI-CV (doubting/checking, obsessing, ordering, and washing) alongside seeking proxies for internal states and sensory sensitivity.

## 3. Results

### 3.1. Demographic Characteristics

The mean age of the parents in our sample was 38.18 (range 26–55), and the mean number of their children was 3.24 (range 1–11). The great majority of parents reported having an academic degree (66%) or a high-school or professional school diploma (27%). Family income was reported as average or above average in 59% of the parents, with the remaining reporting a below-average income. Medical issues of the children were reported by 13.9% of the sample, with 18.1% reporting developmental, emotional, or social issues, and 37.6% reporting previous emotional or physical therapy for their child. Among participating parents, 321 (79.3%) were born in Israel, 22 (5.4%) in the United States, 17 (4.2%) in Russia, and the rest in other countries. 

Descriptive statistics for our main dependent measures are depicted in Table 1, and the correlations between them are depicted in Table 2, including for boys and girls and the father- and mother-report, respectively. As predicted, obsessive–compulsive symptoms (OCI-CV) were positively correlated with anxiety, sensory sensitivity, and seeking proxies for internal states, and this was true in all subsamples. No significant association between age and any of the variables emerged, but girls (M = 22.35 [SD = 8.22]) were reported to experience more difficulties with sensory sensitivity than boys (M = 20.28 [SD = 6.40], t (403) = −2.80, *p* < 0.01, Cohen’s *d* = −0.28).

### 3.2. Linear Regression, Dominance Analysis, and Network Analysis

The results of the linear regression are presented in Table 3. The linear regression model explained 55.4% of the variation in obsessive–compulsive symptoms. Age and gender were not significantly associated with obsessive–compulsive symptoms, but the tendency to seek proxies for internal states, sensory sensitivity, and anxiety symptoms were. The strongest association emerged between the tendency to seek proxies for internal states and obsessive–compulsive symptoms. Very similar results emerged for regression analyses conducted separately for boys and girls and for the father- and mother-report, respectively (the results can be obtained from the corresponding author). Dominance analysis showed that the tendency to seek proxies for internal states explained 29.9% of the unique variance in obsessive–compulsive symptoms, followed by sensory sensitivity (15.9%), anxiety symptoms (10.0%), age (0.1%), and gender (0.0%).

The network of the variables is presented in Figure 2. All variables were uniquely associated with each other, with a particularly strong association emerging between the tendency to seek proxies for internal states and obsessive–compulsive symptoms. We first tested whether the tendency to seek proxies for internal states was more strongly associated with obsessive–compulsive symptoms (edge = 0.51, 95% CI: 0.44–0.58) than with anxiety (edge = 0.25, 95% CI: 0.16–0.34). The posterior probability was 100%, confirming our hypothesis. The prediction that both the tendency to seek proxies for internal states and sensory sensitivity would be uniquely associated with OCD (hypothesis 2) was also confirmed, as both 95% CIs excluded zero (see Figure 2). Finally, the posterior probability for hypothesis 3, that is, that the association between the tendency to seek proxies for internal states and obsessive–compulsive symptoms (edge = 0.51, 95% CI: 0.44–0.58) would be larger than the association between sensory sensitivity and obsessive–compulsive symptoms (edge = 0.25, 95% CI: 0.16–0.34) was 100%, confirming hypothesis 3.

Figure 3 shows the network of the selected subscales of OCI-CV and the other variables. The tendency to seek proxies for internal states was uniquely associated with all obsessive–compulsive symptom dimensions except obsessing. The strongest association emerged between the tendency to seek proxies for internal states and ordering (edge = 0.30, 95% CI: 0.21–0.38). No unique associations emerged between sensory sensitivity and obsessive–compulsive symptoms.

## 4. Discussion

The present study aimed to examine the associations between obsessive–compulsive symptoms, sensory sensitivity, and seeking proxies for internal states among 5 to 10-year-old children, using parental reports. We showed, for the first time, that the tendency to seek proxies for internal states is strongly related to obsessive–compulsive symptoms in this age group. Notably, seeking proxies for internal states was significantly associated with all major obsessive–compulsive symptom dimensions except obsessing. This indicates that the SPIS model is of broad relevance to obsessive–compulsive symptoms in children, particularly to the compulsive aspects of such symptoms. As predicted, seeking proxies for internal states was more strongly related to obsessive–compulsive symptoms than to anxiety, in line with previous evidence that the attenuation of access to the internal states and compensatory reliance on proxies are specific to OCD and not attributable to anxiety (e.g., [43,44]; see [10] for a review).

Also, in accordance with our hypotheses, there was a significant positive correlation between obsessive–compulsive symptoms and sensory sensitivity, corroborating previous findings of associations between abnormalities in sensory sensitivity and pediatric as well as adult obsessive–compulsive symptoms [5,12,13,14,15,16,17,18,19,20,21]. One interpretation of these findings is that children who experience overwhelming sensory input combined with difficulties in accessing the internal states may be at risk of developing obsessive–compulsive behaviors. However, a recent longitudinal study did not find evidence that SOR, as reported by parents in relation to their children, predicted later OCD symptoms [19], so any causal conclusions about this relationship would clearly be immature.

Keeping in mind the preliminary nature of this study, it is nevertheless intriguing to consider the associations documented in this study via the role of shared reality in the development of access to the internal states. Learning to identify our internal states, including our feelings, emotions, and motivations, has been recognized as a major developmental challenge by both early (e.g., [45]) and modern theorists (e.g., [46,47]). Theories of social development agree that empathic reflection and the sharing of internal states (e.g., when a parent says to the child “this must have hurt” or “wasn’t this fun?”) is critical for successfully negotiating this challenge. When such sharing and reflection is hindered, as would likely be the case for children with sensory over-responsiveness, then the ability to access the internal states may not fully develop. According to the SPIS model, such attenuation of access to the internal states would lead children to seek and rely on proxies for these states, including fixed rules and behavioral routines. Adopting such proxies may be a way for the child to circumvent the need to rely on their difficult-to-access internal states. For example, adopting a specific diet and fixed portions of food can bypass having to assess one’s level of hunger and preference for specific foods, both of which are internal states. Notably, a recent review [48] suggests that the age range sampled in our study is an important period in the development of rituals, both those that are part of normal development and those that may be precursors of OCD.

Hypothesizing about such causal processes is clearly speculative; however, exploring them requires further research using non-correlational designs. One avenue for such research would be to apply a factorial design, whereby participants are divided into groups of high/low sensory sensitivity and high/low obsessive–compulsive symptoms. These participants will undergo experimental paradigms developed in previous SPIS model studies, which are designed to demonstrate seeking and relying on proxies in clinical and non-clinical samples (e.g., in the form of real and false biofeedback for muscle tension [43,44,49,50]. Grouping participants by the level of sensory sensitivity and obsessive–compulsive symptoms might allow for a better understanding of the effect of each factor on proxy-seeking, as well the effects of the interaction between these factors. Prospective studies can also help to shed light on the temporal associations between seeking proxies for internal states, sensory sensitivity, and obsessive–compulsive symptoms in children.

In addition to the SPIS model, there are several other potential accounts of the relationships between OCD and SOR in children, as well as in adults. For example, Rossi et al. [20] suggested that OCD, and particularly compulsive behaviors, may be attributable to hypo-functioning of sensory gating in OCD. A later study by Steinman and colleagues [51], however, did not support the hypothesized sensorimotor gating deficit in OCD. Another potential account was put forth by Russo et al. [21], who suggested that OCD is associated with a dysfunction of sensory-motor integration, which may give rise to both SOR and compulsive behaviors. A similar hypothesis was proposed by Van Hulle and colleagues [19]. In addition, these authors hypothesized that some compulsions may arise in response to unpleasant sensations, particularly in children who display SOR. They also noted that these two types of symptoms are moderately heritable and may share genetic influence, so that the development of both may be affected by overlapping genetic risk factors. Clearly, the present results cannot distinguish between these competing (and perhaps complementary) accounts.

The current study has several limitations. First, the study was conducted using parental reports. Alongside its advantages—e.g., easy access to a large population; obtaining fast and direct information; and high internal, content, and face validity—this method is prone to social desirability bias. For example, parents might have exaggerated while answering the survey if they assumed that higher scores would qualify them to participate in future surveys. Furthermore, parental reports are obviously imperfect, as parents have limited access to their children’s inner world and might view or interpret their behavior Inaccurately. Second, several of the questionnaires used in the current research were altered, translated, or developed ad hoc. Although the statistical analyses indicated adequate internal consistency for all questionnaires used, there is a need to test whether these results will replicate in future studies and with different research populations. Third, our sample was composed of children between the ages of five and ten. Although this age range encompasses a crucial period in the development of OCD [52,53], it does not include some important developmental milestones, such as puberty. Furthermore, the restricted age range and correlational nature of the study constrain any conclusions regarding the developmental trajectory of OCD throughout childhood and adolescence. Future research might benefit from longitudinal designs that follow children with obsessive–compulsive symptoms over time and compare participants’ sensory sensitivity and proxy-seeking behavior across different ages. This design would enable researchers to observe any changes in the correlations found in the present study throughout the developmental course of OCD. Fourth, the study examined parents’ reports of children’s obsessive–compulsive symptoms in a non-clinical population. Future studies of clinical populations could provide additional insights regarding the connections between seeking proxies for internal states, sensory sensitivity, and obsessive–compulsive symptoms.

In conclusion, our study found that seeking proxies for internal states is strongly associated with obsessive–compulsive symptoms in children during middle childhood. Importantly, these associations emerged in relation to all major expressions of OCD except for obsessing and after accounting for sensory sensitivity and anxiety symptoms. The present findings may be of importance for understanding the mechanisms underlying the development and/or maintenance of obsessive–compulsive symptoms in children.

## Figures and Tables

**Figure 1 brainsci-13-01463-f001:**
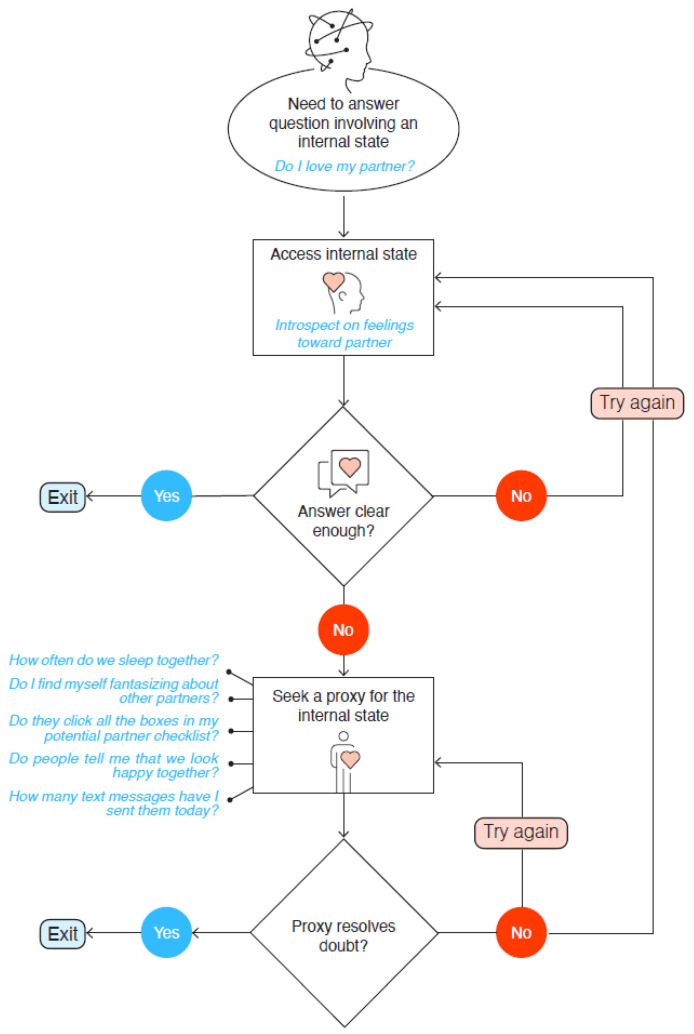
Illustration of the SPIS Model. Note. The process at the core of the SPIS model sets off when a person wants to answer a question about an internal state, such as “do I love my partner?” Accessing this internal state may or may not provide a clear answer. If the answer is clear, the process terminates. If it is not, the person may try to access the internal state again or seek a proxy for it, such as counting the number of text messages exchanged daily. The proxy may or may not resolve the doubt. If the doubt is not resolved, the process is repeated. According to the SPIS model, OCD is characterized by attenuated access to internal states, which increases the likelihood of repeated looping through the process, for example, in the form of compulsions.

**Figure 2 brainsci-13-01463-f002:**
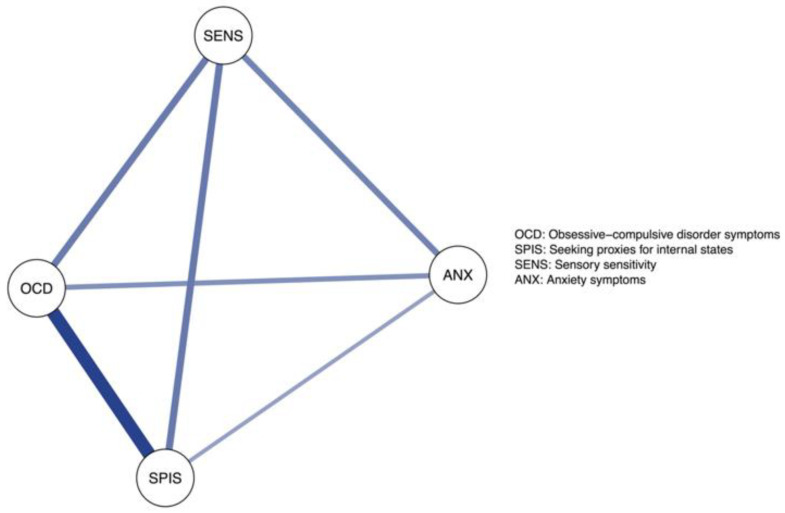
The Network Structure of Obsessive–Compulsive Symptoms, the Tendency to Seek Proxies for Internal States, Sensory Sensitivity, and Anxiety Symptoms.

**Figure 3 brainsci-13-01463-f003:**
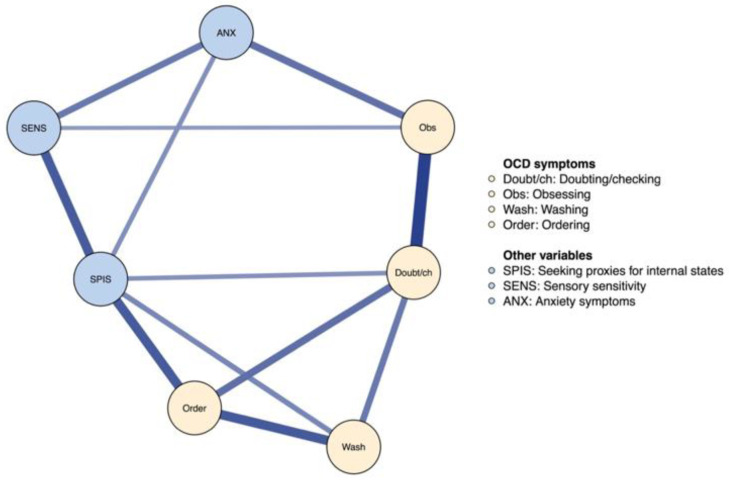
The Network Structure of OCD Symptom Dimensions, the Tendency to Seek Proxies for Internal States, Sensory Sensitivity, and Anxiety Symptoms.

**Table 1 brainsci-13-01463-t001:** Descriptive statistics for study variables.

	M (SD)	Min–Max	Skewness	Kurtosis
SPISI-CV Total Score	41.60 (14.07)	21–95	0.56	−0.13
OTHS Total Score	21.38 (7.49)	12–49	1.02	0.24
OCI-CV Total score	31.44 (10.61)	21–75	1.40	1.93
OCI-CV Doubting/checking	6.80 (2.51)	5–20	1.99	4.72
OCI-CV Obsessing	6.40 (2.81)	4–17	1.39	1.67
OCI-CV Ordering	4.91 (2.55)	3–15	1.50	1.68
OCI-CV Washing	4.33 (2.00)	3–12	1.67	2.28
SCARED Total Score	7.40 (1.99)	5–14	0.99	0.35

**Table 2 brainsci-13-01463-t002:** Correlations between key variables.

Full Sample	OCI-CV	SPISI-CV	OTHS	SCARED
OCI-CV	-	0.71 *	0.59 *	0.49 *
SPISI-CV		-	0.59 *	0.47 *
OTHS			-	0.48 *
SCARED				-
Girls (above diagonal) Boys (below diagonal)	OCI-CV	SPISI-CV	OTHS	SCARED
OCI-CV	-	0.72 *	0.55 *	0.50 *
SPISI-CV	0.68 *	-	0.60 *	0.44 *
OTHS	0.65 *	0.57 *	-	0.49 *
SCARED	0.47 *	0.51 *	0.47 *	-
Mothers (above diagonal) Father (below diagonal)	OCI-CV	SPISI-CV	OTHS	SCARED
OCI-CV	-	0.72 *	0.60 *	0.44 *
SPISI-CV	0.68 *	-	0.59 *	0.44 *
OTHS	0.57 *	0.57 *	-	0.41 *
SCARED	0.54 *	0.50 *	0.56 *	-

* *p* < 0.001.

**Table 3 brainsci-13-01463-t003:** Results from linear regression with OCD symptoms as the dependent variable and seeking proxies for internal states, sensory sensitivity, anxiety symptoms, age, and gender as independent variables. Associations are presented as standardized beta coefficients (βs).

Independent Variable	β	95% Confidence Interval for β	*p*
Seeking proxies for internal states	0.51	0.43–0.59	<0.001
Sensory sensitivity	0.22	0.14–0.31	<0.001
Anxiety symptoms	0.15	0.07–0.22	<0.001
Age	0.03	−0.04–0.09	0.43
Gender	−0.02	−0.08–0.05	0.65

Notes. OCD = Obsessive–compulsive disorder.

## Data Availability

The data presented in this study are available on request from the corresponding author. The data are not publicly available due to its sensitivity.

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
