# Peer review of "Obsessive–Compulsive Symptoms in Children Are Related to Sensory Sensitivity and to Seeking Proxies for Internal States"

_brainsci, 2023, doi:10.3390/brainsci13101463_

Round 1

Reviewer 1 Report

The central idea and focus of the paper are highly innovative and thought-provoking. There is indeed fragmented evidence suggesting a potential link between OCD symptoms and aberrant sensory processing, particularly in a subset of patients. More systematic and model-driven research on this topic is undoubtedly needed, and I would like to commend the authors for pursuing this important goal.

However, a key concern arises from the current state of the paper, which reads somewhat underdeveloped. Testing a promising theory in a somewhat "sloppy" manner and overinterpreting results could potentially do a disservice to the theory, especially if it is relatively recent in its introduction.

Key conceptual points:

1.       Integration of Related Concepts/Results: The conceptual section of the paper could benefit from better integration of related concepts and findings from prior literature. For instance, attentional functions, such as sensory gating, could potentially lead to aberrant sensory processing. There is a significant body of literature suggesting that this process is indeed aberrant in some individuals with OCD, including insights into sexual dimorphism in these impairments. Examples of relevant studies include Rossi (2005, 2014), Javanbakht (2006), Xiao (2010), Steinman (2016), and Ahmari (2016). It's important to recognize how altered attentional processes might impact sensory processing, possibly by overloading the system. In such cases, the issue may not solely concern access to internal states but also revolve around the challenge of integrating and managing sensory overload.

2.       Related Model: Autogenous/Reactive Model of Obsessions: Another closely related model that partially overlaps is the autogenous/reactive model of obsessions (Lee and Kwon, 2003). This model posits that autogenous obsessions (related to a sense of incompleteness, sexual, religious, and mental compulsions) are more strongly linked to sensory deficits than to anxiety. Moreover, these obsessions tend to occur more frequently in younger individuals and exhibit a significant (large effect size) male predominance. Numerous empirical tests of this model have established connections between autogenous obsessions and various sensory difficulties, as well as altered functioning of the salience network.

In summary, there is room for your model to be more effectively situated within the existing literature. Additionally, at this stage, it's crucial to keep track of alternative explanations and to carefully delineate which explanations your new tests reject and which remain feasible. Several alternative hypotheses could emerge, all of which might align with your data. This is especially relevant given that the central scale you employ has not yet been well validated, including in terms of external and discriminatory validity. The symptom of pathological doubt, a hallmark of OCD, could easily overlap with many items on the scale, warranting further research in this area and caution against overinterpretation of the results.

3.       Developmental Component: The developmental aspect could benefit from more thorough exploration, particularly given the targeted special issue. The ages of 5 to 10 are pivotal in the development of OCD, with many children experiencing obsessive-compulsive behaviors that are developmentally appropriate. What is referred to as "access to internal states" could potentially signify attempts to learn how to interpret these internal states. While the paper touches on this process in the context of abnormal development, it would be valuable to also address normal developmental aspects.

Methodological points:

1.       Distribution of OCS and Subthreshold OCs: It would be informative to include information regarding the distributions of OCS (obsessive-compulsive symptoms) and subthreshold OCs in your analysis. Are there any differences in the relationships of interest among subgroups where OCs are present versus subthreshold OCs, and do these differences vary by gender or age? Given the large sample size and the developmental angle, you have the opportunity to conduct more nuanced tests. OCD is highly heterogeneous, and literature suggests that while some individuals with OCs may indeed have sensory difficulties, it might not be the case for all. Consider using group membership (e.g., gender, AO/OR type) as a modulator of the relationships of interest.

2.       Mediation Model: The theoretical part of the paper suggests a mediation model where sensory sensitivity impacts OCS through tendencies to seek external proxies (sensory sensitivity → tendency to seek proxies → obsessive-compulsive symptoms, particularly compulsive behavior). However, the statistical model employed differs from this proposed mediation model. It would be beneficial to clarify the rationale behind this choice. As is, the theoretical model does not match the statistical model.

3.       Relying on Parental Reports: A significant limitation of the study lies in relying on parental reports. Existing literature has shown only moderate agreement between parental and children's reports (e.g., GÜRBÜZ, 2022), with more agreement found on observable/external aspects than on non-observable/internal domains (e.g., Hemmingsson, 2017), which are the focus of your paper. It is crucial to be upfront about this limitation and to discuss it in the abstract, background, and methods sections rather than briefly mentioning it as a limitation at the end.

4.       Potential Biases Related to Parental Characteristics: Some research, such as Roy (2010), has shown that when children report the most symptoms and impact, qualitative aspects of the parent-child relationship and family structure seem to be more influential predictors of disagreement than the gender of the child and socio-demographic variables. When parents report the most symptoms and impact, low parental educational level, low income, and the gender of the child play an additional role. Therefore, it is important to assess potential biases associated with these parental characteristics. It might be worthwhile to evaluate differences in parent-child dyads (e.g., mother-sons, mother-daughters, father-sons, father-daughters) and consider other relevant characteristics.

In summary, while the central idea of your paper is commendable and thought-provoking, addressing these conceptual and methodological points will enhance the clarity, rigor, and overall quality of your research.

Author Response

Reviewer 1

The central idea and focus of the paper are highly innovative and thought-provoking. There is indeed fragmented evidence suggesting a potential link between OCD symptoms and aberrant sensory processing, particularly in a subset of patients. More systematic and model-driven research on this topic is undoubtedly needed, and I would like to commend the authors for pursuing this important goal.

However, a key concern arises from the current state of the paper, which reads somewhat underdeveloped. Testing a promising theory in a somewhat "sloppy" manner and overinterpreting results could potentially do a disservice to the theory, especially if it is relatively recent in its introduction.

Key conceptual points:

  1. Integration of Related Concepts/Results: The conceptual section of the paper could benefit from better integration of related concepts and findings from prior literature. For instance, attentional functions, such as sensory gating, could potentially lead to aberrant sensory processing. There is a significant body of literature suggesting that this process is indeed aberrant in some individuals with OCD, including insights into sexual dimorphism in these impairments. Examples of relevant studies include Rossi (2005, 2014), Javanbakht (2006), Xiao (2010), Steinman (2016), and Ahmari (2016). It's important to recognize how altered attentional processes might impact sensory processing, possibly by overloading the system. In such cases, the issue may not solely concern access to internal states but also revolve around the challenge of integrating and managing sensory overload.

Author Response: We thank the reviewer for pointing out the references that suggest further views on the relationships between sensory abnormalities and OCD. We have now integrated the relevant papers into the discussion (just before the limitation section) and have also added other papers that suggest alternative potential accounts for the relationships between OCD and SOR (e.g., Van Hulle and colleagues).

  1. Related Model: Autogenous/Reactive Model of Obsessions: Another closely related model that partially overlaps is the autogenous/reactive model of obsessions (Lee and Kwon, 2003). This model posits that autogenous obsessions (related to a sense of incompleteness, sexual, religious, and mental compulsions) are more strongly linked to sensory deficits than to anxiety. Moreover, these obsessions tend to occur more frequently in younger individuals and exhibit a significant (large effect size) male predominance. Numerous empirical tests of this model have established connections between autogenous obsessions and various sensory difficulties, as well as altered functioning of the salience network.

Author Response: We are familiar with (and have read again) the cited article and related literature on autogenous obsessions and could not find findings relevant to sensory dysregulation in relation to these type of obsessions. If we had missed something, we would be glad to add the relevant findings to the manuscript.

In summary, there is room for your model to be more effectively situated within the existing literature. Additionally, at this stage, it's crucial to keep track of alternative explanations and to carefully delineate which explanations your new tests reject and which remain feasible. Several alternative hypotheses could emerge, all of which might align with your data. This is especially relevant given that the central scale you employ has not yet been well validated, including in terms of external and discriminatory validity. The symptom of pathological doubt, a hallmark of OCD, could easily overlap with many items on the scale, warranting further research in this area and caution against overinterpretation of the results.

Author Response: We agree with these critical comments, and have toned down the conclusions from our findings, which clearly went far from the actual data. We also discuss other interpretations of our findings, as noted above in the context of our response to point 1.

  1. Developmental Component: The developmental aspect could benefit from more thorough exploration, particularly given the targeted special issue. The ages of 5 to 10 are pivotal in the development of OCD, with many children experiencing obsessive-compulsive behaviors that are developmentally appropriate. What is referred to as "access to internal states" could potentially signify attempts to learn how to interpret these internal states. While the paper touches on this process in the context of abnormal development, it would be valuable to also address normal developmental aspects.

Author Response: We agree, and we did discuss the importance of shared reality and empathy in the development of normal access and recognition of internal states in the Discussion of the original version of the manuscript. We now also added a reference for the finding that the age range sampled in our study is important in the development of rituals, both those are part of normal development and those that may be precursors of OCD.

Methodological points:

  1. Distribution of OCS and Subthreshold OCs: It would be informative to include information regarding the distributions of OCS (obsessive-compulsive symptoms) and subthreshold OCs in your analysis. Are there any differences in the relationships of interest among subgroups where OCs are present versus subthreshold OCs, and do these differences vary by gender or age? Given the large sample size and the developmental angle, you have the opportunity to conduct more nuanced tests. OCD is highly heterogeneous, and literature suggests that while some individuals with OCs may indeed have sensory difficulties, it might not be the case for all. Consider using group membership (e.g., gender, AO/OR type) as a modulator of the relationships of interest.

Author Response: We agree it would be very interesting to examine the frequency of children with elevated OC symptoms.  Unfortunately, we changed the response scale of OCI-CV and cannot use the published cutoff points. Moreover, the established cutoff points have been under criticism, and in our own experience do not really distinguish clinical from subclinical participants (see also work by Amitai Abramovitch and colleagues on this issue). In relation to other potential moderators, in the revised version, we examine whether child age and child sex explain to differences in the key variables of the study and examine associations among the major variables in girls and boys separately and for father- and mother-report, respectively. We do this using both correlations and regression.

  1. Mediation Model: The theoretical part of the paper suggests a mediation model where sensory sensitivity impacts OCS through tendencies to seek external proxies (sensory sensitivity → tendency to seek proxies → obsessive-compulsive symptoms, particularly compulsive behavior). However, the statistical model employed differs from this proposed mediation model. It would be beneficial to clarify the rationale behind this choice. As is, the theoretical model does not match the statistical model.

Author Response: We see your point but given that the data we have access to cannot be used to test true mediation in a valid way, we did not conduct a mediation analysis. We see the paper as a first step in examining the relations between these variables in children more broadly. This has been clarified in the revised introduction. 

  1. Relying on Parental Reports: A significant limitation of the study lies in relying on parental reports. Existing literature has shown only moderate agreement between parental and children's reports (e.g., GÜRBÜZ, 2022), with more agreement found on observable/external aspects than on non-observable/internal domains (e.g., Hemmingsson, 2017), which are the focus of your paper. It is crucial to be upfront about this limitation and to discuss it in the abstract, background, and methods sections rather than briefly mentioning it as a limitation at the end.

Author Response: We agree and now discuss this decision more explicitly in the methods section. We should also note, however, that this method has been used extensively, in our own work as well as in the more recent review by Van Hulle and colleagues which we now cite extensively in our paper. We should also note that we intentionally selected items which are relatively observable and do not require guessing on the side of the parents as to their children's internal states, such as their feelings and beliefs.

  1. Potential Biases Related to Parental Characteristics: Some research, such as Roy (2010), has shown that when children report the most symptoms and impact, qualitative aspects of the parent-child relationship and family structure seem to be more influential predictors of disagreement than the gender of the child and socio-demographic variables. When parents report the most symptoms and impact, low parental educational level, low income, and the gender of the child play an additional role. Therefore, it is important to assess potential biases associated with these parental characteristics. It might be worthwhile to evaluate differences in parent-child dyads (e.g., mother-sons, mother-daughters, father-sons, father-daughters) and consider other relevant characteristics.

In summary, while the central idea of your paper is commendable and thought-provoking, addressing these conceptual and methodological points will enhance the clarity, rigor, and overall quality of your research.

Author Response: Thanks for this interesting suggestion, Accordingly, we have now examined the zero-order associations and regression among the key variables for father and mother report, respectively.

Reviewer 2 Report

The purpose of the study was not clearly stated in the study. I am not sure whether this sentence is the aim, “we aimed to account also for the relationship between SPIS, sensory sensitivity and anxiety symptoms.”. You need to state the need for this study. Any call from previous studies? Or problems needed to be addressed?

What’s the significance of this study? How can it contribute to the existing research field?

Regarding participants, why did only the children need to be 5 to 10 years old? In the literature, you seem didn’t to mention this group of children is the most concern group. Why not 3 to 12 or 6 to 17?

In the discussion, the explanation of the findings of the hypotheses seemed to be weak. More evidence and explanation should be given to support your findings.

Any theoretical implication or practical implication?

The conclusion is a bit too brief. 

Author Response

Reviewer 2

The purpose of the study was not clearly stated in the study. I am not sure whether this sentence is the aim, “we aimed to account also for the relationship between SPIS, sensory sensitivity and anxiety symptoms.”. You need to state the need for this study. Any call from previous studies? Or problems needed to be addressed?

Author Response: We believe that the major contributions of the study are twofold. First, it is the first test of the SPIS model in children. Second, it is an attempt to start examine how SPIS, sensory sensitivity, and obsessive-compulsives symptoms relate to each other. We see the study as a first step towards examining these relations. This has been clarified in last sections of the introduction.

What’s the significance of this study? How can it contribute to the existing research field?

Author Response: As noted above, these are the first findings relating these variables to each other. We believe that it would lead to further research as we suggested in the Discussion.

Regarding participants, why did only the children need to be 5 to 10 years old? In the literature, you seem didn’t to mention this group of children is the most concern group. Why not 3 to 12 or 6 to 17?

Author Response: As we now note in the Introduction, this age group is important in the development of both normal rituals and those that may later develop into OCD.

In the discussion, the explanation of the findings of the hypotheses seemed to be weak. More evidence and explanation should be given to support your findings.

Author Response:  We are not sure why the reviewer thinks that the explanations are weak. At any rate, we now expanded the Discussion of the results and are suggesting other accounts of the findings.

Any theoretical implication or practical implication?

Author Response: Again, we now expand the discussion related to implications of the findings.

The conclusion is a bit too brief. 

Author Response: We wanted the conclusion to be brief so as not to make it redundant with the Abstract and the Discussion. If there is a major point that seems to be missing, we would be glad to add it to the concluding statement.

Reviewer 3 Report

Firstly, I am writing to express my gratitude for the opportunity to review the research article “Obsessive-Compulsive Symptoms in Children are Related to Sensory Sensitivity and to Seeking Proxies for Internal States”. I am honored to have been selected to contribute to the peer-review process for Brain Sciences.

I understand the critical importance of rigorous evaluation in academic research and am eager to lend my expertise to this process. I am confident that my analysis will be of value to the authors and help ensure that the work is of the highest quality.

Thank you for entrusting me with this important task. I look forward to the opportunity to provide a thorough and constructive review.

This study discusses the correlation between symptoms of obsessive-compulsive disorder (OCD) and atypical sensory processing, particularly sensory over-responsivity, in both children and adults. In adults, OCD symptoms are associated with limited access to internal states and a reliance on external proxies such as fixed rules and rituals. The study's objective was to explore the relationship between sensory over-responsivity, the inclination to use proxies for internal states, and OCD symptoms in children aged five to ten. The analysis of data from 404 children's parents showed a stronger association between seeking proxies for internal states and OCD symptoms compared to anxiety symptoms. While sensory over-responsivity was also linked to OCD symptoms, the connection was less prominent than with seeking proxies for internal states. 

I would like to make a series of improvement suggestions to the authors:

INTRODUCTION

Clarity of Definitions:

Define OCD, obsessions, and compulsions more precisely, incorporating the latest diagnostic criteria as per standard psychiatric manuals (e.g., DSM-5 or ICD-10).

Incorporate Recent Statistics:

Include the most up-to-date prevalence rates and statistics related to pediatric OCD to ensure accuracy and relevance.

Detail on Heterogeneity:

Provide a more detailed discussion on the diverse symptom presentations, severity levels, and responses to treatment within pediatric OCD to highlight the true extent of heterogeneity.

Comprehensive Explanation of SPIS Model:

Elaborate on the Seeking Proxies for Internal States (SPIS) model, explaining its theoretical underpinnings, core components, and how it relates to OCD in a clear and comprehensive manner.

Evidence Base for SPIS Model:

Offer a concise overview of the empirical evidence that supports the SPIS model, including key studies, methodologies, and findings that validate its relevance to OCD.

Integration of Previous Studies:

Integrate more recent and relevant studies to provide a comprehensive review of existing literature on the association between obsessive-compulsive symptoms, sensory sensitivity, and seeking proxies for internal states in both children and adults.

Sensory Processing Models:

Discuss other established models of sensory processing to provide a well-rounded understanding of how sensory sensitivity may contribute to obsessive-compulsive symptoms in children.

In-depth Sensory Sensitivity Description:

Delve deeper into sensory sensitivity, explaining its neurobiological underpinnings, how it's measured, and its potential links to OCD, particularly in children.

Citation of Primary Sources:

Ensure accurate citation and reference to primary sources of information, especially when discussing previous studies, theories, or models, to uphold academic rigor and reliability.

METHOD

Participant Demographics:

Provide a more comprehensive description of participant demographics, including socio-economic factors, cultural diversity, and geographic distribution to enhance generalizability and understanding of the sample characteristics.

Sampling Methodology and Representativeness:

Detail the sampling methodology (e.g., random sampling, stratified sampling) and discuss the representativeness of the chosen sample in relation to the population of interest.

Questionnaire Modification Rationale:

Provide a rationale for adapting the OCI-CV scale to a 5-point Likert-type scale, addressing potential impacts on data interpretation and comparability with previous studies.

Psychometric Properties:

Expand on the psychometric properties of the scales used, including additional information on validity, reliability, and any modifications made to the scales for this particular study.

Comprehensive Sensory Sensitivity Measurement:

Incorporate multiple measures to capture a comprehensive assessment of sensory sensitivity, considering including more scales or methods to ensure a thorough evaluation.

Diversity in Anxiety Measurement:

Utilize multiple anxiety measurement scales to capture diverse anxiety dimensions (e.g., panic/somatic anxiety, generalized anxiety) and provide a comprehensive understanding of anxiety symptoms in relation to other variables.

Alternative Statistical Analyses:

Explore and justify the use of alternative statistical analyses (e.g., structural equation modeling, mediation analysis) to provide a more comprehensive analysis of the relationships among variables.

RESULTS

Demographic Characteristics Analysis:

Consider conducting a deeper analysis of demographic characteristics (e.g., age, number of children, education level) to explore potential correlations with the main dependent measures.

Network Analysis Interpretation:

Provide a more thorough interpretation of the network analysis results, emphasizing the most influential connections and the potential implications for understanding the interplay of variables in pediatric OCD.

DISCUSSION

Elaborate on Shared Reality Role:

Provide a more detailed discussion on the role of shared reality in accessing internal states, emphasizing its significance in the context of sensory sensitivity and seeking proxies for internal states.

Propose Experimental Designs:

Suggest specific experimental designs to further investigate causal processes and interactions between sensory sensitivity and obsessive-compulsive symptoms in seeking proxies for internal states.

Advocate for Prospective Studies:

Emphasize the need for prospective studies to elucidate the temporal associations between seeking proxies for internal states, sensory sensitivity, and obsessive-compulsive symptoms in children.

Address Limitations Effectively:

Provide a more comprehensive discussion of the study's limitations, acknowledging potential biases and offering potential strategies to mitigate them in future research.

Discuss Age-Related Considerations:

Delve into the potential implications of the age range limitation and highlight the importance of future research covering a broader age spectrum, potentially through longitudinal designs.

Highlight Implications for OCD Development:

Expand on the potential implications of the study's findings for understanding the development and maintenance of obsessive-compulsive symptoms in children, considering the broader clinical context.

Continue the excellent effort; your commitment and diligence are clearly reflected in the caliber of your research. I am confident that, with some refinements, this manuscript will be prepared for submission.

Reviewing your work has been a delightful experience, and I am assured that, with the proposed revisions, your paper will significantly enrich the field. I extend my best wishes for your ongoing research endeavors and eagerly anticipate your forthcoming publications.

Regards,

Author Response

Reviewer 3

Firstly, I am writing to express my gratitude for the opportunity to review the research article “Obsessive-Compulsive Symptoms in Children are Related to Sensory Sensitivity and to Seeking Proxies for Internal States”. I am honored to have been selected to contribute to the peer-review process for Brain Sciences.

I understand the critical importance of rigorous evaluation in academic research and am eager to lend my expertise to this process. I am confident that my analysis will be of value to the authors and help ensure that the work is of the highest quality.

Thank you for entrusting me with this important task. I look forward to the opportunity to provide a thorough and constructive review.

This study discusses the correlation between symptoms of obsessive-compulsive disorder (OCD) and atypical sensory processing, particularly sensory over-responsivity, in both children and adults. In adults, OCD symptoms are associated with limited access to internal states and a reliance on external proxies such as fixed rules and rituals. The study's objective was to explore the relationship between sensory over-responsivity, the inclination to use proxies for internal states, and OCD symptoms in children aged five to ten. The analysis of data from 404 children's parents showed a stronger association between seeking proxies for internal states and OCD symptoms compared to anxiety symptoms. While sensory over-responsivity was also linked to OCD symptoms, the connection was less prominent than with seeking proxies for internal states. 

I would like to make a series of improvement suggestions to the authors:

INTRODUCTION

Clarity of Definitions:

Define OCD, obsessions, and compulsions more precisely, incorporating the latest diagnostic criteria as per standard psychiatric manuals (e.g., DSM-5 or ICD-10).

Author Response: We agree it is important to clearly define the hallmark features of OCD, but we feel that the description in the paper is in line with the latest diagnostic criteria.

Incorporate Recent Statistics:

Include the most up-to-date prevalence rates and statistics related to pediatric OCD to ensure accuracy and relevance.

Author Response: Unfortunately, the prevalence of pediatric OCD has not been synthesized in a meta-analysis, but the larger literature was narratively described in a recent overview of OCD which we now cite.

Detail on Heterogeneity:

Provide a more detailed discussion on the diverse symptom presentations, severity levels, and responses to treatment within pediatric OCD to highlight the true extent of heterogeneity.

Author Response: The symptom heterogeneity of pediatric OCD has been described in the introduction of the revised manuscript.

Comprehensive Explanation of SPIS Model:

Elaborate on the Seeking Proxies for Internal States (SPIS) model, explaining its theoretical underpinnings, core components, and how it relates to OCD in a clear and comprehensive manner.

Author Response: We believe that we did that. If the reviewer finds that something specific is missing in our description we would be glad to clarify it.

Evidence Base for SPIS Model:

Offer a concise overview of the empirical evidence that supports the SPIS model, including key studies, methodologies, and findings that validate its relevance to OCD.

Author Response: We included as a reference a review of the SPIS model which describes in detail both its theoretical underpinnings as well as the supporting evidence. We also included a figure that describes the process hypothesized by the SPIS mode. We do not think there is room in the current manuscript to get into more specific details.

Integration of Previous Studies:

Integrate more recent and relevant studies to provide a comprehensive review of existing literature on the association between obsessive-compulsive symptoms, sensory sensitivity, and seeking proxies for internal states in both children and adults.

Author Response: We have now added more recent studies that examined atypical sensory processing in youth with OCD, both in the Introduction and in the Discussion. There are no new studies that involve the SPIS in this context – this was the goal of the present study.

Sensory Processing Models:

Discuss other established models of sensory processing to provide a well-rounded understanding of how sensory sensitivity may contribute to obsessive-compulsive symptoms in children.

Author Response: We have done that, and now include more relevant research on this topic.

In-depth Sensory Sensitivity Description:

Delve deeper into sensory sensitivity, explaining its neurobiological underpinnings, how it's measured, and its potential links to OCD, particularly in children.

Author Response: We have added relevant research on this topic, including theories that suggest potential neurobiological basis that are common to both OCD and sensory sensitivity.

Citation of Primary Sources:

Ensure accurate citation and reference to primary sources of information, especially when discussing previous studies, theories, or models, to uphold academic rigor and reliability.

Author Response: We believe we have done that, but if the reviewer can point out specific errors, we would be glad to correct them.

METHOD

Participant Demographics:

Provide a more comprehensive description of participant demographics, including socio-economic factors, cultural diversity, and geographic distribution to enhance generalizability and understanding of the sample characteristics.

Author Response: We believe that our initial presentation of sociodemographic factors is sufficient as it includes age, number of children, education, income, and medical issues in children. In the revised version, we have added country of birth of the parent. These are the data we collected, so it is hard for us to report more.

Sampling Methodology and Representativeness:

Detail the sampling methodology (e.g., random sampling, stratified sampling) and discuss the representativeness of the chosen sample in relation to the population of interest.

Author Response: As we describe in the methods section, there is no sampling in these types of survey studies. It is open to any participants in the panel who fit the selection criterial and are interested in participating.

Questionnaire Modification Rationale:

Provide a rationale for adapting the OCI-CV scale to a 5-point Likert-type scale, addressing potential impacts on data interpretation and comparability with previous studies.

Author Response: We explained the rationale in terms of making it easier for the responders. This means of course that means and SDs of the modified scales cannot be compared to studies that used the standard tools, but would have no effect on all the other analyses (correlations, regression, network analysis).

Psychometric Properties:

Expand on the psychometric properties of the scales used, including additional information on validity, reliability, and any modifications made to the scales for this particular study.

Author Response: We believe that we have provided all this information and are not sure what exactly is missing. Again, if the reviewer believes something specific should be added we would be glad to do that.

Comprehensive Sensory Sensitivity Measurement:

Incorporate multiple measures to capture a comprehensive assessment of sensory sensitivity, considering including more scales or methods to ensure a thorough evaluation.

Author Response: The study is already conducted, and it is impossible for us to include more/new scales.

Diversity in Anxiety Measurement:

Utilize multiple anxiety measurement scales to capture diverse anxiety dimensions (e.g., panic/somatic anxiety, generalized anxiety) and provide a comprehensive understanding of anxiety symptoms in relation to other variables.

Author Response: The study is already conducted, and it is impossible for us to include more scales. We believe a short measure of anxiety symptoms is motivated in the specific study.

Alternative Statistical Analyses:

Explore and justify the use of alternative statistical analyses (e.g., structural equation modeling, mediation analysis) to provide a more comprehensive analysis of the relationships among variables.

Author Response: We do not believe that SEM is justified. It is just a special case of regression and dominance analysis, for example, suits our needs better as it estimates degree of explained variance of each independent variable. A mediation analysis would be in line with our proposed model, but cross-sectional data of the kind we have access to cannot be used to validly test mediation. Thus, we take a more exploratory stance and simply present the cruder relations between the variables.

RESULTS

Demographic Characteristics Analysis:

Consider conducting a deeper analysis of demographic characteristics (e.g., age, number of children, education level) to explore potential correlations with the main dependent measures.

Author Response: We have added results examining correlations between age and each of the major variables of the study as well as an analysis examining potential sex differences. No significant associations/differences emerged.

Network Analysis Interpretation:

Provide a more thorough interpretation of the network analysis results, emphasizing the most influential connections and the potential implications for understanding the interplay of variables in pediatric OCD.

Author Response: We do not want to overinterpret the network results. We see it as an effective and conservative way to compare unique associations among variables.

DISCUSSION

Elaborate on Shared Reality Role:

Provide a more detailed discussion on the role of shared reality in accessing internal states, emphasizing its significance in the context of sensory sensitivity and seeking proxies for internal states.

Author Response: We have done that.

Propose Experimental Designs:

Suggest specific experimental designs to further investigate causal processes and interactions between sensory sensitivity and obsessive-compulsive symptoms in seeking proxies for internal states.

Author Response: We have done that to the best of our ability, suggesting further studies that can examine the hypotheses put forth in the present research.

Advocate for Prospective Studies:

Emphasize the need for prospective studies to elucidate the temporal associations between seeking proxies for internal states, sensory sensitivity, and obsessive-compulsive symptoms in children.

Author Response: Again, we believe we have done that.

Address Limitations Effectively:

Provide a more comprehensive discussion of the study's limitations, acknowledging potential biases and offering potential strategies to mitigate them in future research.

Author Response: We have included a comprehensive limitation section.

Discuss Age-Related Considerations:

Delve into the potential implications of the age range limitation and highlight the importance of future research covering a broader age spectrum, potentially through longitudinal designs.

Author Response: We specifically suggested longitudinal studies in our Discussion, and are now citing one study that has used this methodology (Van Hulle et al.).

Highlight Implications for OCD Development:

Expand on the potential implications of the study's findings for understanding the development and maintenance of obsessive-compulsive symptoms in children, considering the broader clinical context.

Author Response: We have added many references to the developmental literature related to childhood OCD in the Introduction and Discussion.

Continue the excellent effort; your commitment and diligence are clearly reflected in the caliber of your research. I am confident that, with some refinements, this manuscript will be prepared for submission.

Reviewing your work has been a delightful experience, and I am assured that, with the proposed revisions, your paper will significantly enrich the field. I extend my best wishes for your ongoing research endeavors and eagerly anticipate your forthcoming publications.

Round 2

Reviewer 2 Report

Thank you very much for revising it and addressing my concerns regarding your work. I think it is satisfactory.